# See, Hear, Explore: Curiosity via Audio-Visual Association

**Victoria Dean**
Carnegie Mellon University
vdean@cs.cmu.edu

**Shubham Tulsiani**
Facebook AI Research
shubhtuls@fb.com

**Abhinav Gupta**
Carnegie Mellon University
Facebook AI Research
abhinavg@cs.cmu.edu

## Abstract

Exploration is one of the core challenges in reinforcement learning. A common formulation of curiosity-driven exploration uses the difference between the real future and the future predicted by a learned model [1]. However, predicting the future is an inherently difficult task which can be ill-posed in the face of stochasticity. In this paper, we introduce an alternative form of curiosity that rewards novel associations between different senses. Our approach exploits multiple modalities to provide a stronger signal for more efficient exploration. Our method is inspired by the fact that, for humans, both sight and sound play a critical role in exploration. We present results on several Atari environments and Habitat (a photorealistic navigation simulator), showing the benefits of using an audio-visual association model for intrinsically guiding learning agents in the absence of external rewards. For videos and code, see https://vdean.github.io/audio-curiosity.html.

## 1 Introduction

Many successes in reinforcement learning (RL) have come from agents maximizing a provided extrinsic reward such as a game score. However, in real-world settings, reward functions are hard to formulate and require significant human engineering. On the other hand, humans explore the world driven by intrinsic motivation, such as curiosity, often in the absence of rewards. But what is curiosity and how would one formulate it?

Recent work in RL [1–3] has focused on curiosity using future prediction. In this formulation, an exploration policy receives rewards for actions that lead to differences between the real future and the future predicted by a forward dynamics model. In turn, the dynamics model improves as it learns from novel states. While the core idea behind this curiosity formulation is simple, putting it into practice is quite challenging. Learning and modeling forward dynamics

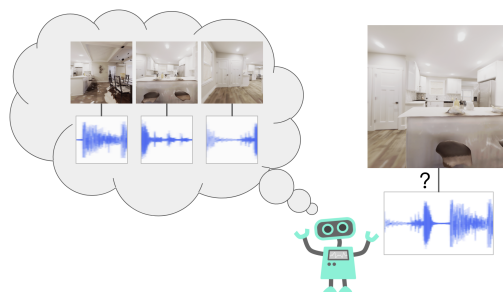

Figure 1: **See, Hear, Explore**: We propose a formulation of curiosity that encourages the agent to explore novel associations between modalities, such as audio and vision. In Habitat, shown above, our method allows for more efficient exploration than baselines.

is still an open research problem; it is unclear how to handle multiple possible futures, whether to explicitly incorporate physics, or even what the right prediction space is (pixel space or some latent space).

The use of multiple modalities in *human* learning has a long history. Research in psychology has suggested that humans look for incongruity [4]. A baby might hit an object to hear what it sounds like. Have you ever found yourself curious to touch a material different from anything you have seen before? Humans are drawn towards discovering and exploring novel associations between different modalities. Dember and Earl [5] argued that intrinsic motivation arises with discrepancy between expected sensory perception and the actual stimulus. More recent work has shown the presence of multimodal stimulation and exploration in infants [6, 7]. In cognitive development, both sight and sound guide exploration: babies are drawn towards colorful toys that squeak and rattle [8].

Inspired by human exploration, we introduce See Hear Explore (SHE): a curiosity for novel associations between sensory modalities (Figure 1). SHE rewards actions that generate novel associations (shared information) between different sensory modalities (in our case, pixels and sounds). We first demonstrate that our formulation is useful in several Atari games: SHE allows for more exploration, is more sample-efficient, and is more robust to noise compared to existing curiosity baselines on these environments. Finally, we show experiments on area exploration in the realistic Habitat simulator [9]. Our results demonstrate that in this setting our approach significantly outperforms baselines.

To summarize, our contributions in this paper include: 1) SHE, a curiosity formulation that searches for novel associations in the world. To the best of our knowledge, multimodal associations have not been investigated in self-supervised exploration; 2) we show our approach outperforms the commonly-used curiosity approaches on standard Atari benchmark tasks; 3) most importantly, multimodality is one of the most basic facets of our rich physical world (audio and vision are generated by the same physical processes [10]). We show experiments on realistic area exploration in which SHE significantly outperforms baselines. This work builds on efficient exploration, which will be crucial as we push agents to explore more complex unknown environments.

## 2   Related Work

Our work uses audio as an additional modality for self-supervised exploration. We divide the prior work into two categories: exploration (Section 2.1) and multimodal learning (Section 2.2).

### 2.1   Exploration

Prior work on exploration has used error [11, 12, 1, 13], uncertainty [14, 15, 3], and potential improvement [16] of a prediction model as intrinsic motivation. Some approaches have used count-based or pseudo-count-based exploration [17, 18]. Others use auxiliary losses to supplement reward functions and improve sample efficiency [19, 20].

One popular approach to self-supervised exploration is curiosity by self-supervised prediction [1, 2]. In this form of curiosity, an intrinsic reward encourages an agent to explore situations with high error under a jointly-trained future prediction model. The model's error is a proxy for novelty: unpredictable situations are more likely novel and therefore ones the agent should explore. These future-predicting models can be difficult to train, especially in visual space. Our method also looks at self-supervised exploration, but our intrinsic reward does not rely on future prediction. We circumvent the need for predicting the future by leveraging multimodal input. SHE rewards association classification error (i.e. association novelty) as opposed to higher-dimensional prediction error. Our key insight is that *associative* models across modalities are simpler to learn, and their accuracy is also indicative of novelty.

### 2.2   Multimodal Learning

Multimodal settings are especially amenable to self-supervision, as information from one modality can be used to supervise learning for another modality. One prior work learned a joint visual and language representation using Flickr images and associated descriptors [21]. In computer vision, audio can provide additional information that complements images [22–24]. Recent work [25, 26] has looked at audio-visual embodied navigation, in which audio is emitted from a goal point to aid in supervised learning of navigation. In the same environment, Gao et al. [27] used audio and visual information for learning visual feature representations. We test on this audio-visual navigation environment, but for unsupervised exploration in RL; we have no goal states.

Audio and visual information are closely linked, and since we commonly have access to both in the form of video, this is a rich area for self-supervision. Aytar et al. [28] used audio from Atari in the form of YouTube videos of people playing the games. This work uses audio-visual demonstrations from YouTube to learn a visual embedding. The setup here is learning from demonstrations from humans. In our case, on the other hand, the audio-visual associations drive intrinsically motivated exploration. We learn multimodal alignment from active data, which the agent both collects and uses.

In robotics settings, the use of additional modalities such as tactile sensing [29, 30] or audio [31] is increasingly popular for grasping and manipulation tasks. Lee et al. [32] showed the effectiveness of self-supervised training of tactile and visual representations by demonstrating its use on a peg insertion task. While these previous approaches have demonstrated the benefits of using multiple sensory modalities for learning better representations or accurately solving tasks, in this work we demonstrate its utility for allowing agents to explore. To the best of our knowledge, using audio to learn actions for exploration is unique to our work.

## 3    See, Hear, Explore

We now describe SHE, our exploration method based on associating audio and visual information. Our goal is to develop a form of curiosity that exploits the multimodal nature of the input data. Our core idea is that the SHE agent learns a model that captures associations between two modalities. We use this model to reward actions that lead to unseen associations between the modalities. By rewarding such actions, we guide the exploration policy towards discovering new combinations of sight and sound.

More formally, we consider an agent interacting with an environment that contains visual and sound features, which we call $x_t = (v_t, s_t)$ for time $t$ where $v_t$ is the visual feature vector and $s_t$ is the sound feature vector. The agent explores using a policy $a_t \sim \pi(v_t; \theta)$ where $a_t$ corresponds to an action taken by the agent at time $t$. To make for easier comparison to visual-only baselines, our agent is only given access to the visual features $v_t$ and not the audio features $s_t$. To enable this agent to explore, we train a discriminator D that tries to determine whether an observed multimodal pair $(v_t, s_t)$ is novel, and we reward the agent in states where the discriminator is surprised by the observed multimodal association.

### 3.1    Why Novel Associations?

The goal of an exploration policy is to perform actions that uncover states that lead to a better understanding of the world. One commonly used exploration strategy involves rewarding actions that lead to unseen or novel states [17]. While this strategy seems intuitive, it does not handle the fact that while some states might not have been seen, we still understand them and hence they do not need to be explored. In light of this, recent approaches have used a prediction-based formulation. If a model cannot predict the future, it needs more data points to learn. However, sometimes we may have seen enough examples, and prediction is still challenging, leading a prediction-based exploration policy to get stuck. For example, consider the couch-potato issue: the random TV in the Unity environment (as described in Burda et al. [2]) yields high error for prediction models, so prediction-based curious agents receive high rewards for staring at the TV, though this is not a desirable type of exploration.

Trying to avoid these problems has shaped much of the work on intrinsic motivation; Schmidhuber [16], Oudeyer et al. [33], White et al. [34], and Burda et al. [35] all formulate intrinsic rewards with the goal of mitigating problems like the couch-potato agent. Our approach, different from this body of prior work, looks at how multimodal data can mitigate these issues.

Our underlying hypothesis is that discovering new sight and sound associations will help mitigate the shortcomings of the previously described count-based and prediction-based exploration strategies. By using an association model, we ask a simpler question: can this image co-occur with this sound? Consider another example, in which pressing a button randomly produces one among 3 distinct sounds. Our approach could learn to classify all as associated, while an agent using future prediction error would always be curious. This focus on association effectively helps ignore stochasticity, mitigating the couch-potato problem by focusing on non-random structure. Such a model can allow generalization to unseen states, and it also does not need to predict the future to provide an informative signal for exploration.

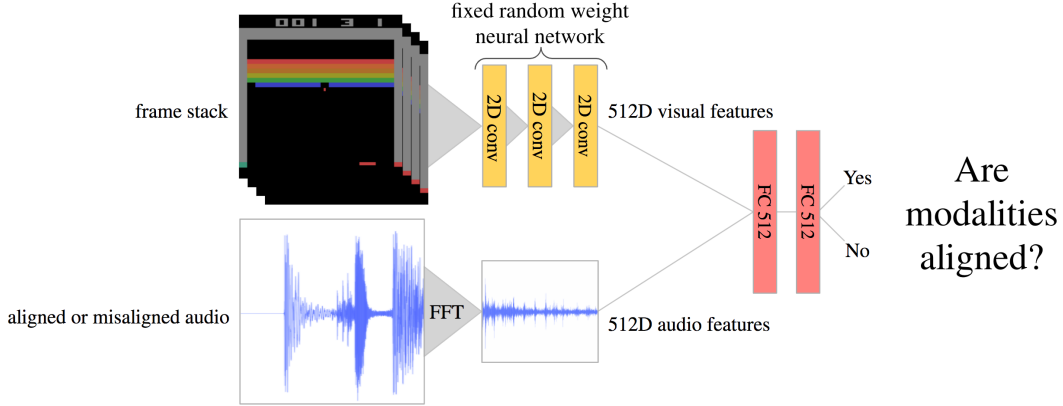

Figure 2: **Our audio-visual association model**: The frames (top left) and potentially misaligned audio waveform (bottom left) are preprocessed into 512-dimensional feature vectors using a random feature network and FFT, respectively. The discriminator network (right) takes these features as inputs and is trained to output whether or not they are aligned. 2D conv represents a standard convolutional layer and FC 512 represents a fully-connected layer with 512 units.

## 3.2 Association Novelty via Alignment

The core of our method is the ability to determine whether a given pair $(v_t, s_t)$ represents a novel association. To tackle this problem, we learn a model in an online manner. Given past trajectories, a model learns whether a certain audio-visual input comes from a seen or new phenomenon. One way to model this would be to use a generative model such as a VAE [36] or GAN [37], which could determine if the image-audio combination is within the distribution or out of distribution. However, generative models are also difficult to train, so we instead propose using a discriminator to predict if the image-audio pair is novel, which has a much smaller, binary output space.

We train this discriminator to distinguish real audio-visual pairs from 'fake' pairs from another distribution, with the insight that the learned model is more likely to classify novel pairs as fake. Here, the observed image-audio pairs during exploration act as positive training examples, but a critical question is how to obtain negative image-audio pairs. To this end, we reformulate the problem as whether image-audio pairs are aligned or not: we obtain 'fake' samples by randomly misaligning the audio and visual modalities, similar to Owens and Efros [23]. The positive data is then the aligned image-audio pairs, and the negative data is comprised of misaligned ones. The discriminator model, as shown in Figure 2, outputs values between 0 and 1, with 1 representing high probability of audio-visual alignment and 0 representing misalignment. We can then leverage the misalignment likelihood as an indicator of novelty since the discriminator would be uncertain in such instances.

## 3.3 Training

Having introduced association novelty via alignment, we now describe how we implement this idea using function approximators. During training, the agent policy is rolled out in parallel environments. These yield trajectories which are each chunked into 128 time steps. A trajectory consists of pairs of preprocessed visual and sound features: $(v_1, s_1), (v_2, s_2)...(v_{128}, s_{128})$. These trajectories are used for two purposes: 1) updating the discriminator D as described below and 2) updating the exploration policy based on the intrinsic reward $r_t^i$ (computed using the discriminator), also described below.

**Training the Alignment Discriminator** The discriminator D is a neural network that takes a visual and sound feature pair as input and outputs an alignment probability. To train D, we start with positive examples from the visual and sound feature pairs $(v_t, s_t)$. With 0.5 probability we use the true aligned pair, and with 0.5 probability we create a false pair consisting of the true visual feature vector $v_t$ and a sound feature vector uniformly sampled from the current trajectory. We call this false sound $s_t'$. We define a binary variable $z_t$ to indicate whether the true audio was used, i.e. when we give the discriminator the true audio $s_t$, we set $z_t = 1$, and when we give the discriminator the false audio $s_t'$,

$z_t = 0$. We use a cross-entropy loss to train the discriminator, similar to prior work [23, 28]:

$$\mathcal{L}_t(v_t, s_t, z_t) = \begin{cases} -\log(D(v_t, s_t)), & \text{if } z_t = 1 \\ -\dfrac{||s_t - s'_t||_2}{\mathbb{E}_{\text{batch}}||s_t - s'_t||_2} \log(1 - D(v_t, s'_t)), & \text{if } z_t = 0 \end{cases}$$

In the $z_t = 0$ case above, we weight the cross-entropy loss to prevent the discriminator from being penalized in cases where the true and false audio are similar. We weight by the L2 difference between the true and false audio feature vectors and normalize by dividing by the mean difference across samples in the batch of 128 trajectories. This loss is used for updating the discriminator and is not used in computing the agent's intrinsic reward.

**Training the Agent via Intrinsic Reward**    We want to reward actions that lead to unseen image-audio pairs. For a given image-audio pair, if the discriminator predicts 0 (unseen or unaligned), we want to reward the agent. On the other hand, if the discriminator correctly outputs 1 on a true pair, the agent receives no reward. Mathematically, the agent's intrinsic reward is the negative log-likelihood of the discriminator evaluated on the true pairs: $r_t^i := -\log(D(v_t, s_t))$, where the output of D is between 0 and 1. Audio-visual pairs that the discriminator knows to be aligned get a reward of 0, but if the discriminator is uncertain (the association surprised the discriminator) the agent receives a positive reward. The agent takes an action and receives a new observation $v_t$ and intrinsic reward $r_t^i$ (note that the agent does not have access to the sound $s_t$). The agent is trained using PPO [38] to maximize the expected reward: $\max_\theta \mathbb{E}_{\pi(v_t;\theta)} \left[ \sum_t \gamma^t r_t^i \right]$. *The agent does not have access to the extrinsic reward. Extrinsic reward is used only for evaluation.* This will enable the use of our method on future tasks for which we cannot easily obtain a reward function. See Burda et al. [2] for further discussion on training with no extrinsic reward while using it for evaluation.

## 4  Experiments

In this section, we will test our method in two exploration settings (Atari and Habitat) and compare it with commonly-used curiosity formulations.

### 4.1  Environments

**Atari**    Similar to prior work, we demonstrate the effectiveness of our approach on 12 Atari games. We chose a subset of the Atari games to represent environments used in prior work and a range of difficulty levels. We excluded some games due to lack of audio (e.g. Amidar, Pong) or the presence of background music (e.g. RoadRunner, Super Mario Bros). The action space is different from the one used in the future prediction curiosity work [2], as we use Gym Retro [39] in order to access game audio, and Retro environments use a larger action space. The original work reported results using the minimal action space, Discrete(4), whereas we use Discrete(6). We note that the larger action space does slow exploration, but it is used for both our method and the baselines for fair comparisons. To compute audio features, we take an audio clip spanning 4 time steps (1/15th of a second for these 60 frame per second environments) and apply a Fast Fourier Transform (FFT). The FFT output is downsampled using max pooling to a 512-dimensional feature vector, which is used as input to the discriminator along with a 512-dimensional visual feature vector.

**Habitat Navigation**    We also test our method in a navigation setting using Habitat [9] (Figure 3). In this environment, the agent moves around a photorealistic Replica scene [40]. We use the largest Replica scene, Apartment 0, which has 211 discrete locations. In each location, the agent can face in 4 directions. At each timestep, the agent takes one of 3 discrete actions: turn left, turn right, or move forward. As in our Atari experiments, the agent is not given any extrinsic reward; we simply want to see how well it can explore the area without supervision. We use the audio-visual navigation extension from Chen et al. [25], which emits a fixed audio clip from a fixed location and allows our agent to hear the sound after simulating room acoustics. The perceived sound at each time step is less than 1 second long, and we zero pad this audio to 1 second to make each sound equal length for feature computation. We apply FFT and downsample to a 512-dimensional feature vector, the same as done in Atari, described above.

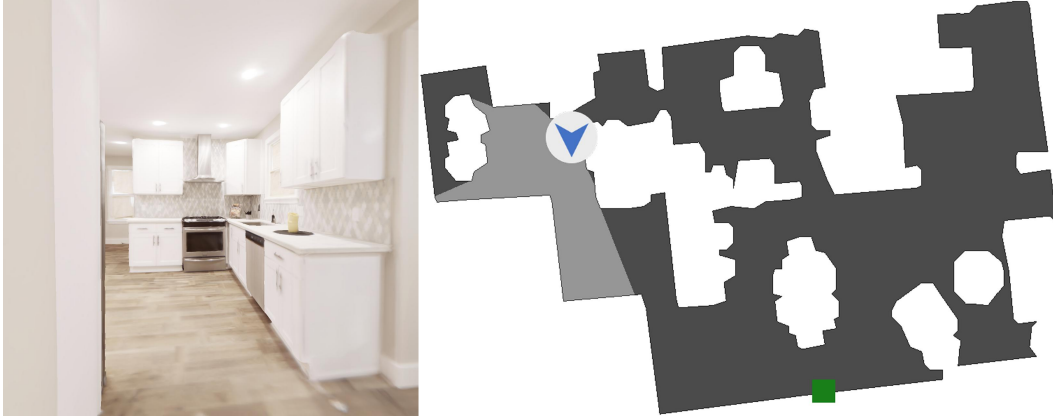

Figure 3: **Habitat visualization**: Left: an example agent view. Right: the top-down map for apartment 0 (not seen by agent). The agent is the blue arrow and the audio source is the green square. Gray areas are open space, while white areas are obstacles, which make exploration challenging.

## 4.2 Baselines

We compare to future prediction curiosity [2], which as previously described performs visual future prediction. We build upon the open-source code from the authors (see the appendix for more details). We also compare to exploration via disagreement [3] and Random Network Distillation (RND) [35]. We use the same hyperparameters (which were optimized for the future prediction and disagreement baselines) for policy learning across all approaches. We use random CNN features [2, 35] for the visual feature representation for our method and the baselines in all experiments.

## 4.3 Atari Experimental Results

We trained our approach and baselines for 200 million frames using the *intrinsic* reward and measure performance by the *extrinsic* reward throughout learning. Figure 4 shows these results. Each method was run with three random seeds, and the plots show the mean and standard error for each method. Please see the appendix for more experimental details. Across many environments, our method enables better exploration (as judged by the extrinsic reward) and is more sample efficient than the baselines. Of the 12 environments, SHE outperforms the disagreement baseline in 9 and the future prediction baseline in 8. We hypothesize that states leading to novel audio-visual associations, such as a new sound when killing an enemy, are more indicative of a significant event than ones inducing high prediction error (which can happen due to inaccurate modeling or stochasticity) and this is why our approach is more efficient across these environments.

**Understanding Failure Cases** While our approach generally exceeds the performance of or is comparable to the curiosity baselines, there are some environments where SHE underperforms. We have analyzed these games and found common failure cases: 1) Audio-visual association is trivial. For example in Qbert, the discriminator easily learns the associations: every time the Qbert agent jumps to any cube the same sound is made, thus making the discriminator's job easy, leading to a low agent reward. Visiting states with already learned audio-visual pairs is necessary for achieving a high score, even though they may not be crucial for exploration. The game Atlantis had similarly high discriminator performance and low agent rewards. 2) The game has repetitive background sounds. Games like SpaceInvaders and BeamRider have background sounds at a fixed time interval, but it is hard to visually associate these sounds. Here the discriminator has trouble learning basic cases, so the agent is unmotivated to further explore. In Alien, the agent quickly learns that by quickly passing from one side of the screen to the other, a sound occurs with a slight delay that makes it hard to align with the frame. The agent learns to repeat this trick continuously, putting the discriminator in a situation similar to 2).

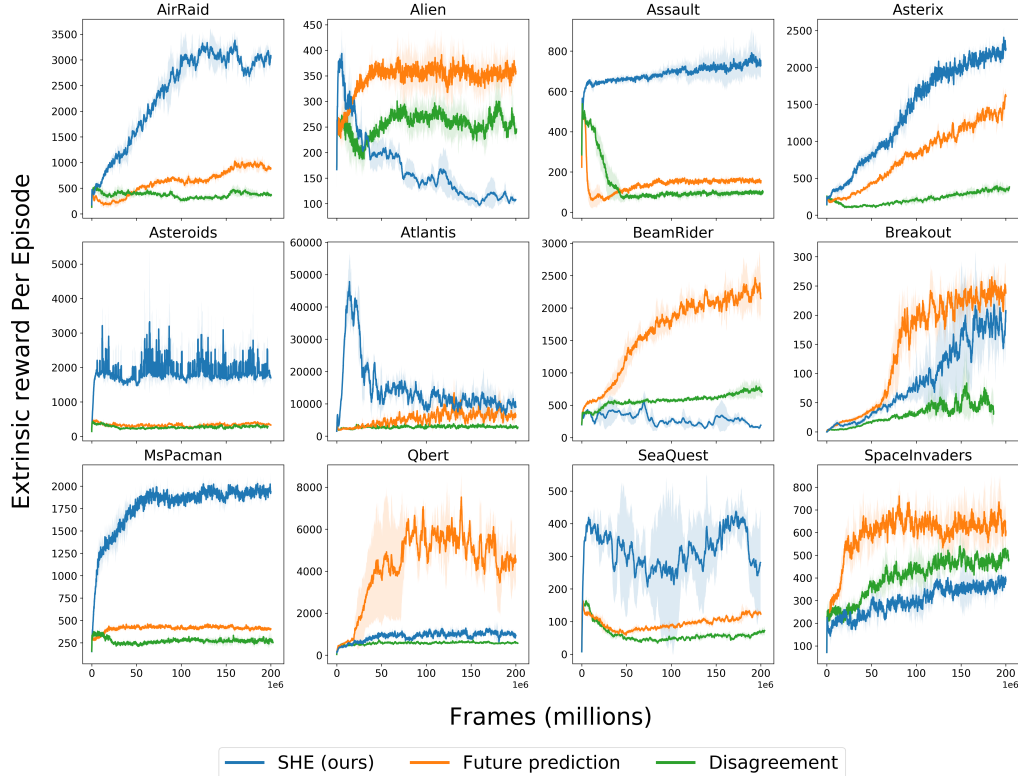

Figure 4: **Atari training curves**: Average extrinsic reward (never seen by the agent) throughout training for our method, future prediction [2], and exploration via disagreement [3]. Our method outperforms the baselines in 8 of 12 environments, supporting our hypothesis that audio-visual association is a useful signal for accelerating exploration.

**Hard Exploration Environment** According to Taïga et al. [41], Gravitar is a hard exploration environment. Such environments are particularly difficult to solve without learning from demonstrations [28], using extrinsic reward [41], or exploiting structure in the game [42]. Even for humans, it can be unclear how to play Gravitar upon first introduction, in contrast with other Atari games that are intuitively simple. Despite Gravitar's difficulty, SHE allows the agent to explore well, while the baselines perform poorly (Figure 5). After examining the game, we hypothesize that the game's visual dynamics are not that interesting on their own, but the audio-visual associations are. We also applied our method and the baselines to other hard exploration games, but in these cases, no method was successful in the training time allotted.

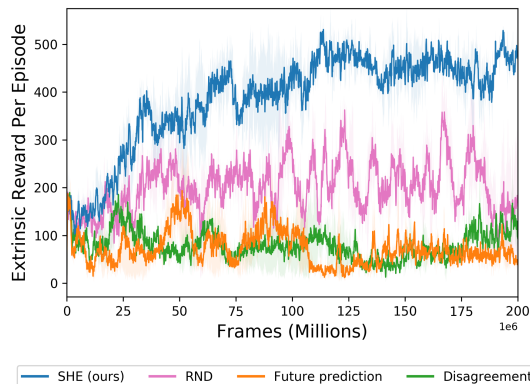

Figure 5: **Case study on Gravitar**: Our method is able to explore this hard environment, while baselines have negligible increase in extrinsic rewards.

## 4.4 Habitat Experimental Results

Here we present results from unsupervised area exploration in the biggest scene in Replica [40] with realistic acoustic responses [25]. Figure 6 shows the quantitative results. SHE (blue) has similar coverage to RND and reaches full state coverage 3 times faster than future prediction curiosity (Figure

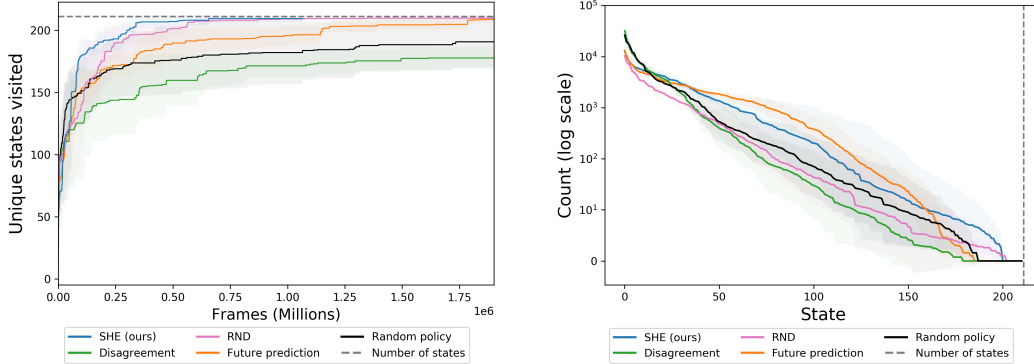

(a) State coverage, as measured by unique states visited throughout training. Our method achieves full state coverage about 3 times faster than future prediction curiosity.

(b) State counts: the number of times each state is visited in the first 2000 episodes, sorted by frequency and shown on a log scale. Our method has a wider tail, visiting rare states one to two orders of magnitude more frequently than the baselines.

Figure 6: **Habitat exploration results for SHE and baselines**. Each method is run with three different seeds and each seed uses a different start location.

6(a)). We can also look at how much each state is visited (Figure 6(b)). A good exploration method will have higher counts in the rare states. Our method visits these rare states (Figure 6(b) right) about 8 times more frequently than the next-best baseline. It does so by visiting common states (Figure 6(b) left) less frequently. SHE's strong performance on this more realistic task holds promise for future work exploring the real world.

## 4.5 Ablations

**Audio in baseline** One hypothesis for why our method outperforms baselines is that SHE has access to additional information in the form of audio. To test the benefit of including audio without the use of our association method, we created two additional baselines: an audio-visual prediction baseline and an audio-visual random network distillation baseline. In the audio-visual prediction baseline, the prediction space is concatenated audio and visual features: the future prediction model takes an audio-visual feature vector as input and predicts an audio-visual feature vector. Similarly, in the audio-visual random network distillation baseline, the audio and visual features are concatenated and used as inputs to both the random target network and the predictor network. As the results in the appendix indicate, this does not lead to significant improvement over the visual-only baselines.

**Robustness to noise** Predicting the future can be especially difficult in the face of inherent uncertainty. To analyze our approach in such a setting, we created a noisy version of the environments, where Gaussian noise is added to the audio and visual feature vector inputs. Our approach can be affected by noise in both audio and visual observations, whereas the baseline is only affected by the visual noise. For these experiments, we chose three environments: one where our method was better (MsPacman), one where the baseline was better (SpaceInvaders), and one where both methods performed well (Asterix). Figure 7 depicts results across these three environments both with and without noise. We observe that future prediction curiosity is not robust to such noise: the performance degrades significantly in both Asterix and SpaceInvaders. In contrast, as our approach only relies on associations, it is more robust to such noise.

**Multiple Curiosity Modules** Curiosity can have multiple forms, e.g. prediction-based and multimodal, and these are complementary to each other. To demonstrate this, we ran a joint method combining intrinsic rewards: we sum the losses from future prediction and the audio-visual discriminator. The resulting method is better than the visual-only baseline in 10 of 12 games, sometimes surpassing both (see the appendix for the detailed results).

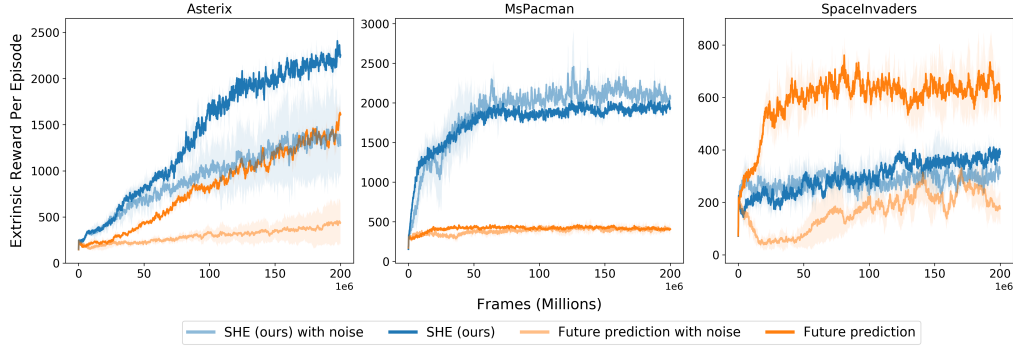

Figure 7: **Effect of input noise on performance**: Our method (blue) maintains similar performance with the introduction of noisy observations, while the baseline performance (orange) degrades.

# 5   Conclusion

Multimodality is one of the most basic facets of our rich physical world. Our formulation of curiosity enables an autonomous agent to efficiently explore a new environment by exploiting relationships between sensory modalities. With results on Atari games, we demonstrated the benefit of using audio-visual association to compute the intrinsic reward. Our method showed improved exploration over baselines in several environments. The most promise lies in our approach's significant gains when used on a more realistic task, exploration in the Habitat environment, where audio and visual are governed by the same physical processes. We anticipate multimodal agents exploring in the real world and discovering even more interesting associations. Instead of building robots that perform like adults, we should build robots that can learn the way babies do. These robots will be able to explore autonomously in real-world, unstructured environments.

## Broader Impact

The lasting impact of RL will be from these algorithms working in the real world. As such, our work is centered around increasing sample efficiency and adaptability. By leveraging self-supervision, we can avoid cumbersome reward shaping, which becomes exponentially more difficult as tasks grow more complex. Although our work here uses simulated agents, our longer-term goal is to deploy multimodal curiosity on physical robots, enabling them to explore in a more sample-efficient manner. Multimodal learning could have a near-immediate impact in autonomous driving, where different sensory modalities are used for perception of near, far, small, and large entities.

Autonomous RL agents have many potential positive outcomes, such as home robots aiding elderly people or those with disabilities. They will save time and money in many sectors of industry. However, they also have the potential to displace parts of the workforce [43].

There could be privacy concerns if merged multimodal data is hard to anonymize or de-identify. There could also be privacy concerns with respect to recording audio data in the wild [44]. With unsupervised RL, it can be hard to predict what behaviors will be learned. For example, a robot using our algorithm might learn to damage sensors to create novel associations. The inability to predict agent behavior can make ensuring safety difficult, which would have consequences in safety-critical settings like autonomous driving or healthcare. Some work has been done on safety in RL [45], and there is more to be done, especially on analyzing the safety of RL exploration policies during training.

## Acknowledgements

The Carnegie Mellon effort was supported by ONR MURI, DARPA MCS, and the ONR Young Investigator Award.

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
