[Supplementary Material]

# A  Code

Our code is available at: `https://github.com/vdean/audio-curiosity`

We use the following open-source repositories for baselines:
Future prediction code: `https://github.com/openai/large-scale-curiosity`
Disagreement code: `https://github.com/pathak22/exploration-by-disagreement`
RND code: `https://github.com/openai/random-network-distillation`

# B  Implementation Details

**Input preprocessing**   We convert all images to grayscale and resize to 84x84. We stack the 4 most recent frames, leading to an observation shape of 84x84x4. In Atari experiments, we use a frameskip that repeats each action 4 times. In Atari, the audio input is 530 samples per timestep, so with the frameskip and frame stack, the audio input is $530 * 4 = 2120$. In Habitat, the audio is 66150 samples per time step (each audio clip is padded to 1 second).

**Network architectures**   The visual embedding network is the same as that used in large scale curiosity [2]. It is a convolutional neural network with fixed random weights. It has 3 fully convolutional layers and a 512-dimensional output. The discriminator network consists of 2 densely connected layers, each with 512 hidden units, followed by another densely connected layer with a single output neuron, representing the likelihood of alignment.

**Hyperparameters**   In Atari, we use 128 parallel environments, and in Habitat, we use 1 environment, as it does not support multithreading. We use the same hyperparameters as in large scale curiosity: a learning rate of 0.0001 for all models, a discount factor $\gamma$ of 0.99, and 3 optimization epochs per rollout.

# C  Combining Multiple Forms of Curiosity

Figure 8: **Complementary forms of curiosity**: Maximizing the sum of SHE and future prediction intrinsic rewards leads to better exploration than future prediction alone on 10 of 12 environments.

Future prediction and multimodal association can be complementary forms of curiosity. To demonstrate this, we present a joint method in which the agent receives combined intrinsic rewards: we sum the losses from future prediction and the audio-visual discriminator. We show results using this joint method in Figure 8.

The resulting combined method is better than the future prediction baseline in 10 of 12 games, sometimes surpassing both (e.g. an average final reward of 593 compared with 108 for SHE and 359 for future prediction in Alien). Further work could explore other ways of combining intrinsic rewards, such as switching between the complementary forms. However, our goal in this work was to introduce our new approach, rather than to show strict performance improvements on every environment by tuning a combined method.

## D    Using Audio in Baselines

Here we present results on using audio in baselines, as described in the main paper ablations section. In the first baseline, the prediction space is concatenated audio and visual features: the intrinsic model takes an audio-visual feature vector as input and predicts an audio-visual feature vector as output. The results from the audio-visual prediction baseline are shown in Figure 9.

Figure 9: **Prediction baseline with audio**: The audio-visual prediction baseline performs similarly to the visual-only baseline, showing that association, rather than simple multimodality, is the key to our method's success.

In the second baseline, we add audio to random network distillation [35]. The audio and visual features are fed through separate neural network layers (dense and convolutional, respectively), concatenated, and used in both the random target network and the predictor network. The results from the audio-visual random network distillation baseline are shown on Gravitar in Figure 10.

These results demonstrate that our method's success is not just because it has additional environment information (audio). The differing sparsities and magnitudes of audio and visual features make it difficult to use both in this way. Meanwhile, our association-based method works well without additional tuning or scaling. As described in the paper, learning association instead of prediction is effective for exploration.

Figure 10: **Random Network Distillation baseline with audio**: On Gravitar, the audio-visual RND baseline performs similarly to the visual-only RND baseline.

# E  Atari with Sticky Actions

While in the main Atari experiments (Section 4.3) we test on similar environments to Burda et al. [35], subsequent works such as Pathak et al. [3] have advocated for the use of a stochastic version of these environments. To test whether our method yields similar benefits when faced with the addition of stochasticity, we ran experiments using sticky versions of each environment.

Figure 11: **Performance on stochastic environments**: In the face of action randomness, our association-based curiosity shows similar performance to the deterministic experiments.

In these "sticky" environments, the agent's action is "stuck" and repeated with probability 0.25 at each time step. These non-deterministic environments can be more challenging for the prediction-based curiosity, as the environment's added randomness makes it harder to predict. Our results, as shown in Figure 11, indicate similar trends to those observed in the non-sticky environments.

# F  Habitat Exploration Visualizations

In the main paper, we presented quantitive results on the Habitat navigation setting. Qualitatively, we can look at heatmaps for where the Habitat agents explore in the scene (Figure 12). We see that baseline agents have poorer visitation in some areas far from where the agent starts (start location marked by arrows). The SHE agent has more uniform coverage, demonstrated by the more consistent shading throughout the map.

(a) SHE (ours)

(b) Future prediction

(c) Disagreement

(d) RND

(e) Random policy

Figure 12: **Navigation heatmaps for our method and the baselines**. Darker represents a higher visitation rate of the state. Visitation rates are computed using the first 2000 training episodes. Our method has more uniform visitation: it visits rarer states more and closer states less, representing better exploration. Heatmaps are made using episodes from one training seed, in which the agent started at the location marked by the arrow.