[Reviews · NeurIPS 2020]

Review 1

Summary and Contributions: This paper presents a self-supervised exploration algorithm for guiding an agent to explore in an environment without external rewards. The curiosity-driven intrinsic reward is generated for novel associations between visual and audio senses. The authors show experiments in two simulated environments: Atari and Habitat. The main contribution is combining visual and audio information of detecting novel areas for exploration.

Strengths: The idea of associating visual and audio information for novelty exploration is interesting.

Weaknesses: - The model for detecting novel association is by evaluating whether the visual and audio signals are aligned. The negative samples are obtained by misaligning paired signals. It can not deal with random TV or objects seen from different angles which result in different visual features -- this causes the seen pair be classified as unseen. For example, P3L119-121, the same reason of random TV causes a prediction-based agent to staring at the TV, will results in the same behavior of the proposed method due to random visual feature and audio feature which make them not aligned and be classified as novel. - The setting is very artificial and only works when there is only one object and one sound at a time and the object is visible when the sound is heard. The proposed method is not applicable to most real scenarios where object is not visible when its sound is heard, multiple objects make the sound at the same time, audio is absent (which is true in most cases) or there are too many audio signals/background noises.

Correctness: P2L72: Although the authors show that the proposed method is better than predictive model-based approach because they claim that the associative model is easier to learn than predictive model in the given setting. In fact both models are difficult to learn, especially the associative model in a more realistic environment where there are multiple sound sources which are difficult to distinguish and to associate with visual objects and the extracted visual features are different when the object is viewed from different angles.

Clarity: The paper is well written and easy to understand.

Relation to Prior Work: It discussed how this work differs from previous contributions.

Reproducibility: No

Additional Feedback: I cannot view the supplementary material. The paper would be stronger if it provides some real scenario experiments. [After rebuttal]: Thanks for providing me the instruction to view the supplementary material. After reading the other reviewer's comments and authors' feedback, I would like to raise my score. The authors have addressed common concerns of all reviewers on formulation of the rewards and explained that the proposed reward can partially solve the couch-potato issue. It's an interesting approach although I think the contributions and applications to real-world scenarios remain limited.


Review 2

Summary and Contributions: This paper introduces a form of multimodal intrinsic motivation, in which the "world model" must determine whether an audio-visual pair sequence is aligned or shifted. It presents experiments that show clear utility over relevant baselines. --------------------------- Update: Thanks to the authors for a thoughtful response! I in particular appreciated the discussion of the error signal as reward, a topic near and dear to my (and it looks like, the other reviewers as well) heart. I'd still like to push them a bit on this, though: while I agree that the alignment problem might be less susceptible to stochasticity, I still think that the world model problem might suffer from a degenerate ill-posedness in a more generalized sense. Consider for instance a 3d environment in which an agent only has partial observability, and for instance, that which is making sound is not in view. This would I think make for its own "white noise". Hence I think it would be great to follow up with experiments in environments with stronger partial observability. As such I don't think this is a *clear* accept, but I still quite appreciate the work.

Strengths: I think that the idea itself is a strong combination of natural and novel, making it very relevant to the NeurIPS community. To my knowledge this is the first multimodal curiosity work. The results are very encouraging, as reasonable baselines have been run.

Weaknesses: One mentioned primary motivation for this work is in trying to improve over future prediction-based curiosity given its susceptibility to unpredictable situations. As noted in the experiments section, the proposed method is susceptible to situations in which audio/visual match is ambiguous -- I would think of this as largely belonging to the same family of vulnerabilities, and more interesting/important to stress-test for than e.g. the gaussian noise applied in one case. In particular, while there might not be so many ambiguities in many atari games, I feel like more life-like environments (e.g. where sound sources are not in view) are completely pervasive and would need to be dealt with carefully. I am glad that the paper includes an ablation in which baselines also receive audio input, but as I think this is a really important point (their model gets more information than many of the baselines!), I wish this were pushed on more. Unless I am misreading, only one baseline is used for this section (visual + audio prediction), but e.g. RND could easily be augmented. As intrigued as I am by the current results, I would be *really* intrigued if it beat that version of RND -- it generally seems like a quite robust baseline! Maybe this goes into the clarity subheader, but I think the consequence is important here -- which baseline from Burda et al ("Large Scale Study...") is used? Pixel prediction? If so, I think that's a bad baseline, and they have several better alternatives. I think *exactly* how audio might be integrated into those alternatives could have a bearing, as well, so more details would be good.

Correctness: Looks good!

Clarity: Yes! Though as noted above, unless I'm missing something there's the crucial bit of clarity on the Burda et al work.

Relation to Prior Work: Yes

Reproducibility: Yes

Additional Feedback:


Review 3

Summary and Contributions: This paper introduces See, Hear, Explore (SHE), a formulation of an intrinsic reward to encourage exploration based on learning what sounds and sights (auditory and visual observations) occur at the same time. Within the method, the agent makes use of a discriminator (within this paper, implemented as a neural network) that learns to recognize what pairs of auditory observations and visual observations occur at the same time in the agent's environment. The discriminator is trained by feeding it both observed audio-visual pairs, labeled as real, and observed audio-visual pairs that have been doctored to be temporally misaligned, labeled as fake. The agent's behaviour is determined by a policy that is, at the same time as the discriminator is being trained, being optimized to maximize the expected return in terms of an intrinsic reward. The intrinsic reward at a given time step is based on the certainty of the discriminator about whether the present (real) audio-visual observation is real. The more certain that the discriminator is that the observation is false, the more novel the observation is thought to be, so the higher the intrinsic reward received by the agent. The paper also presents experimental results considering performance in 12 Atari environments and state coverage in the Habitat simulator. The results are competitive and promising.

Strengths: I enjoyed the overall idea of this paper; the authors share a story following some intuitive inspiration to some promising results. The description of the proposed method and experimental setup are acceptably detailed and clear. The experiments demonstrate that the method appears to have potential to improve exploration in some settings. The paper includes some discussion of the limitations of the work and hypotheses about why these limitations are observed.

Weaknesses: My biggest concern with this paper is the treatment of error as reward, or as this paper refers to it, "curiosity by self-supervised prediction." The "couch-potato" issues associated with using error as reward (described in lines 117-121) have been known for decades (e.g., Schmidhuber, 1991, towards the end of Section 3) yet we seem to have to keep re-discovering them. Can you address why it makes sense to use error as reward in your setting despite this problem? It seems particularly concerning since a stated "longer-term goal is to deploy multimodal curiosity on physical robots," a setting with inherent stochasticity. Could you please provide some reasons why you believe that "discovering new sight and sound associations" (lines 122-123) could mitigate the couch-potato problem? It seems clear that failure case 2 is a version of the couch-potato issues, which suggests that they are not mitigated. The connection between the failure cases and the couch-potato problems should be made clearer and readers should be referred to earlier discussion of the problem. Because the superficial benefits of using error as reward are so tantalizing, I'm concerned about misleading the reader with only the superficial benefits of the approach as it is described in lines 22-34 and 66-73. Will you be able to remedy this concern through revision of your paper? UPDATE (after rebuttal): Thank you for the examples and explanation that seem to indicate that this method might potentially avoid some cases of the couch-potato problem. In particular, the paper would benefit from discussing which instances of the couch potato problem it manages to avoid and how, and which ones remain. As I mentioned previously, the section on Understanding Failure Cases comes closest to answering the second question. The discussion in the rebuttal is a good start towards answering the first question but needs to be expanded upon. I think a very important contribution of this paper could be to help readers understand these questions and the authors should think carefully about how to achieve this. Based on the authors' intentions to deal with the concerns I raised in my initial review, I am raising my score. Since I am unable to see the changes made, I cannot strongly recommend acceptance, but I am sufficiently confident that the paper is above the acceptance threshold.

Correctness: To the best of my knowledge, most of the claims, methods, and empirical methodology are correct. I have a few fairly minor concerns. Lines 39-40: "Dember and Earl [5] argued that intrinsically motivated behaviors tend to create a discrepancy between expected sensory perception and the actual stimulus." This sentence doesn't seem quite right. I think that Dember and Earl actually argued that experimenters have been able to arouse intrinsically motivated behaviours by creating discrepancies between expectation and observation, so it would be good to rephrase the sentence. Line 277: "thus only resulting in a smaller performance drop in one environment." I'm not sure the performance drop is convincingly smaller than the performance drop shown by visual curiosity, nor am I sure that comment really needs to be made.

Clarity: For the most part, the paper is acceptably clear (aside from the concerns I mentioned under Weaknesses). One important concern I had was that the word 'association' appears to be a cornerstone of this work, but it is never clearly defined. Do you mean a recognition that two things are sometimes cotemporal, or is there more to it? I think the first use of the word 'association' where I really want to have intuition about what you mean is in line 44, where you are starting to describe how SHE works. I do have a number of other, more minor questions stemming from statements I found somewhat ambiguous and some suggestions for improving clarity. Lines 199-200: "The perceived sound at each time step is less than 1 second long, and we zero pad this audio to 1 second." I didn't really understand this; is there some connection between a time step and second in this setting, or is this done to make each sound of equal length, perhaps for feature representation reasons? Line 206: "(which were optimized for baselines)" Can you clarify which baselines? Specifically forward-dynamics/visual curiosity, RND, and disagreement? If so you could probably make it clearer by saying, "(which were optimized for these baselines)." Lines 220-230: It would be helpful to see which games fell under each case listed. For example what happened in Alien? Lines 265, 270: The term "intrinsic model" is new, and needs a definition, or could be taken out and replaced with something more precise. I'm guessing that it means the forward dynamics model for visual prediction curiosity and the discriminator for SHE. Is the RL agent receiving the noisy input too?

Relation to Prior Work: The discussion of the prior work is largely functional and provides the information needed for an acceptable understanding of the context of this work. I have a few questions and comments about the discussion of prior work and suggestions to improve its clarity. As part of the motivation for this work, there is a statement that one hypothesis about this design is that it will help mitigate the shortcomings of count-based and prediction-based strategies (lines 122-123), yet there is little discussion of the previous attempts at this goal. Given that trying to avoid these problems has shaped much of the work on computational intrinsic motivation, it might be helpful to see this method in the context of previous attempts. Examples that come immediately to mind are those of Schmidhuber (1991), Oudeyer et al. (2007), and White et al. (2015), who all formulated their intrinsic rewards with the specific goal of avoiding problems like the aforementioned couch-potato issues. Line 22: While I know the meaning of this statement is dependent on one's definition of 'recent,' I would feel more confident if this statement read "Some recent work in RL …" In the past ten years there have been other formulations (for example, consider Still & Precup, 2012). Update to this note: I was reading further down and realized I would also be satisfied with "Recent work in RL [1-3] …" because then I know you're speaking specifically about those references. Lines 35-42: I was somewhat wary of the discussion of prior work within this paragraph because two of the three sources cited are not terribly recent (1965, 1957). I've observed a concerning tendency in computational intrinsic motivation papers to cite seminal works from the psychological sciences without awareness of how understanding of psychological concepts have changed over time amongst psychology researchers. I'm not familiar with this particular area, but I hope that the authors of this work are and that the statements made are representative of current understanding. Lines 174-175: "The agent does not have access to the extrinsic reward. Extrinsic reward is used only for evaluation." This idea is probably going to be weird for anyone who is not familiar with the literature that does this, so a reference to the discussion provided by Burda et al. (2019) about training with no extrinsic reward in a setting evaluated with extrinsic reward, and maybe to other papers where this choice is made, if you have them, would be helpful.

Reproducibility: No

Additional Feedback: Line 64: I was surprised you didn't include Random Network Distillation in this section and I was curious about why not. Line 71: "but our intrinsic reward does not rely on visual prediction." Maybe instead of "visual prediction" in line 71, use the term "future prediction" like you did in line 23. Because you put "visual" on the front of prediction here, it really primed me to think that you were going to do similar predictions, but not of visuals, and it took me perhaps longer than it should have to rid myself of that impression. 'Prediction' is one of those words that has become a bit muddy in the machine learning field, and is sometimes used for the output of a neural network. In that sense, one could say this paper uses predictive models. However, here, when you're saying you aren't using prediction, you mean in the more sense that you aren't making predictions about the future. "We circumvent the need for predictive models by leveraging multimodal input." Many of your readers will be used to the output of a neural network being called a prediction, so I suggest clearly stating whether you allow that meaning or not. Line 87: "associations" being between related sounds and visuals? Figure 2: Can you include an explanation of the FC acronym and conv abbreviation? Line 193: "scene [34]t." Line 203: While I understood what you meant, it felt a little weird to see "large-scale curiosity" used as a name, particularly since I would not consider the curiosity itself to be large-scale. In the referenced paper (Burda et al., 2018, p. 2) they call it "the dynamics-based curiosity model of intrinsic reward presented in Pathak et al. (2017)" but I have also seen the method called Intrinsic Curiosity Module (ICM) (Taïga, 2019, p. 2). Figure 6 caption: "about 3 faster" Line 270: Just to clarify, is the noise being added component-wise to the feature vector? I understand the feature vector to be the concatenation of a visual component, generated through a "random features" method, and an auditory component, generated through a FFT; hopefully that's correct. Line 319: "[4] JMcVLevine Hunt. Intrinsic motivation and its role in psychological development. In Nebraska symposium on motivation, volume 13, pages 189–282. University of Nebraska Press, 1965." (lines 319-320) Not sure what went wrong with the citation here, but I believe the author should be J. McV. Hunt and the editor is David Levine. Also, don't forget to go through and check the capitalization on your references. :) I think the obvious question after reading about this method is, is there any reason that the associations should be between sounds and visuals? I'm imagining if you randomized the order of your (v_t, s_t) vector and used the first half the same way you use v_t now, and take the other half and use it the same way you use s_t right now. What kind of performance would result? I'm not expecting you to answer that question in this paper, but I hope you think about answering it in the future if you develop this method further. In particular, this method might have more general use than the writing of this paper gives it credit for, and experimenting to find out could be quite interesting. Burda, Y., Edwards, H., Pathak, D., Storkey, A., Darrell, T., & Efros, A. A. (2018). Large-scale study of curiosity-driven learning. arXiv preprint arXiv:1808.04355. Burda, Y., Edwards, H., Pathak, D., Storkey, A., Darrell, T., & Efros, A. A. (2019a). Large-Scale Study of Curiosity-Driven Learning. In International Conference on Learning Representations. Burda, Y., Edwards, H., Storkey, A., & Klimov, O. (2019b). Exploration by random network distillation. In International Conference on Learning Representations. Oudeyer, P. Y., Kaplan, F., & Hafner, V. V. (2007). Intrinsic motivation systems for autonomous mental development. IEEE transactions on evolutionary computation, 11(2), 265-286. Pathak, D., Agrawal, P., Efros, A. A., & Darrell, T. (2017, August). Curiosity-driven exploration by self-supervised prediction. In Proceedings of the 34th International Conference on Machine Learning-Volume 70 (pp. 2778-2787). Taïga, A. A., Fedus, W., Machado, M. C., Courville, A., & Bellemare, M. G. (2019). Benchmarking bonus-based exploration methods on the arcade learning environment. In Exploration in Reinforcement Learning Workshop at the 36th International Conference on Machine Learning, Long Beach, California. Schmidhuber, J. (1991). Curious model-building control systems. In Proc. International Joint Conference on Neural Networks (pp. 1458-1463). Still, S., & Precup, D. (2012). An information-theoretic approach to curiosity-driven reinforcement learning. Theory in Biosciences, 131(3), 139-148. White, A., Modayil, J., & Sutton, R. S. (2014, June). Surprise and curiosity for big data robotics. In Workshops at the Twenty-Eighth AAAI Conference on Artificial Intelligence. ADDITION (after rebuttal): After considering your discussion of how your method avoids cases of the couch potato problem, I think this work might be an interesting case of intrinsic reward via compression progress because being able to associate audio and visual signals indicates that the observation as a whole is more compressible. There is considerable work exploring this idea, and I thought you might be interested in thinking about the connections. I am not arguing that these connections need to be included in this particular paper, to be clear.


Review 4

Summary and Contributions: This paper proposes a method to exploit audio-visual correspondence in an RL environment to intrinsically guide agent to explore new states. A discriminator is trained to discriminate whether video states and audio clips are true matches or not. For training the discriminator, positive pairs are in sync and negative are off-sync meaning that the audio-clip is taken from the current trajectory but different time step. The discriminator's prediction is used as an intrinsic reward during the training of the agent - i.e. the agent receives a reward when the predictor guesses incorrectly which can indicate that a previously unexplored - or at least unexpected - state has been visited. The authors show that using this intrinsic reward improves the performance of the agent compared to baselines on the majority of Atari games considered as well as on the Habitat Simulator. Moreover the method is demonstrated to be more robust to visual noise.

Strengths: + Exploiting the audio-visual correspondences in the RL setting seems novel. Multimodal cues are a great source of supervision and critical for learning in humans. It would be great to see more of this line of work. + Experimental results are overall convincing. The proposed method seems to be outperforming the baselines clearly on most experiments. + Relating hard-to-associate video/audio combinations with unexplored/surprising states is a good insight. + The use of the audio cues to aid exploration rather than as part of the state or for representation learning seems novel. + Weighting the cross-entropy loss so that the model is not penalized for not discriminating unsuitable negatives (although no ablation on this, see weaknesses) + Good intuition provided for why using audio-visual associations rather than prediction can lead to better exploration (L.216) + I like the way that the off-synced negatives are sampled from the same trajectory, making the task harder for the discriminator. + Noise ablation is interesting

Weaknesses: 1. Perhaps the most important limitation I can see is the artificial environments used. In games and especially those old Atari ones, audio events can be repeated exaclty the same and it's quite easy for the network to learn to distinguish new sounds, whereas this might not be the case in more realistic environments, where there's more variance and noise in the audio. 2. L.225: "Visiting states with already learned audio-visual pairs is necessary for achieving a high score, even though they may not be crucial for exploration" So that seems like an important limitation, agent won't work well in this sort of environments, which can easily happen in realistic scenarios. 3. L.227 "The game has repetitive background sounds. Games like SpaceInvaders and BeamRider have background sounds at a fixed time interval, but it is hard to visually associate these sounds" Same here, repetitive background sounds might often be the case in real applications. 4. L190: It's a bit strange how the authors use vanilla FFT instead of the more common STFT (overlapping segments and a Hann windowing function). Probably a good idea to try this for consistency with literature. Insufficient ablations: 6. An ablation on the weighting method of the cross-entropy loss would be nice to see. The authors note for example that in Atlantis their method underperforms because "the game has repetitive background sounds". This is a scenario I'd expect the weighting might have helped remedy. 7. An ablation with adding noise to the audio channel would be interesting. 8. An ablation sampling the negatives from unrelated trajectories would also be interesting 9. Some architectural details are missing and some unclear. For example why is the 2D convnet shown in Fig. 2 fixed to random initialization?

Correctness: Yes the experimental sections seems fairly straightforward and well executed.

Clarity: The paper is very well written, easy to read and understand. There is one part that is not clear to me: 9. L. 210: Are the agents first pretrained with the intrinsic rewards and then finetuned with the extrinsic ones, or are they receiving both simultaneously for all the 200 million iterations?

Relation to Prior Work: The authors seem to adequately discuss related work and outline their contributions (I am however not very familiar with literature on exploration for RL). The one other paper that seems to be most relevant [25] by using audio, seems to be using it to aid representation learning, while the authors make it clear that they use audio as intrinsic motivation to guide exploration. Also the difference from vision/speech applications is outlined, as this paper is not about learning an audio-visual task per-se or representation learning, but about using the extra modality to encourage the agent to explore.

Reproducibility: No

Additional Feedback: Some further clarifications would be nice: 10. L.189 "we take an audio clip spanning 4 time steps" What's are the games' frame rates? 11. Fig 6: How come random is so good at coverage? 12. Is the cross-entropy weighting a contribution or have the authors seen something similar in the audio-visual learning literature (in the latter case please provide reference) Suggestions: L.216: "outperforms disagreement in 9 and visual prediction in 8" Maybe make the baseline names italic so it's clear that the words refer to methods. typos / grammar: Fig.6 a) caption "about 3 faster" --> "about 3 times faster" ####### POST REBUTTAL ######## I thank the authors for their feedback on the reviews. Regarding my main concern about generality, the authors make a decent point that they have taken a step towrads more realistic environments, using Habitat. However my concern that the synthetic sounds used in the considered settings are quite non-realistic and make the problem much easier, persists. Moreover It seems that the other reviewers are pointing out the 'couch potato issue' with using the error as a reward. As far as the stocasticity of the future predictions is concerned, I agree with the authors' response that their proposed method is more robust to ambiguity compared to deterministically predicting the future. But I would not be able to tell to what extent this solves the problem and it does not seem to me that they are formally addressing the issue. For all the above, I am lowering my score a bit, to marginally above the acceptance threshold. I still find that the paper overall has sufficiently novel ideas and merits acceptance.

[Author Response · NeurIPS 2020]

We thank the reviewers for their time and feedback. In this paper, we argue that intrinsic motivation can take many
forms. Inspired by humans, we propose a curiosity formulation based on multimodal association: searching for novel
associations to explore. We demonstrate that audio-visual curiosity shows promising results not only on standard Atari
environments but also on the realistic Habitat setting.

We highlight that the reviewers believe the paper is interesting (R1), novel (R4), "the first multimodal curiosity work"
(R2), and shares "intuitive inspiration and some promising results" (R3). There are concerns with regards to the
applicability of the approach and the failure cases, and R3 has concerns with respect to the formulation (specifically the
use of error as reward). We address all of these concerns below. However, the main contribution of our paper is the
introduction of multimodal curiosity. Independent of the specific formulation, this should be of great interest to the
NeurIPS community and pave the way for exploiting the richness of data for better performance.

**R1, R3: Formulation (R3: error as reward, R1: association is as hard, R1,3: couch-potato issue)**: Error as
reward is not necessarily bad; the question is what error. Our approach is not to predict audio given visual features (or
vice-versa). Instead, given both audio and visual features, we classify whether they are aligned or not. We highlight that
this association classification error (i.e. association novelty) is fundamentally different from the prediction error based
formulation typically used. In particular, this has two implications:
(a) In the prior visual curiosity framework, the model predicts the future frame in high-dimensional space. We argue
that compared to higher-dimensional prediction, our classification formulation is less susceptible to the issues
mentioned. For example, if pressing a button produces three distinct sounds, we could learn to classify all of these
as associated, while an agent using future prediction error would always be curious.
(b) With error as reward, as Schmidhuber points out, "the problem is that in non-deterministic environments the
controller will focus on parts of the environmental dynamics which are inherently unpredictable." On the other
hand, in our case, the discriminator focuses on deterministic aspects to solve the alignment classification problem.
This effectively helps ignore stochasticity! Therefore, while our approach would not overcome a purely random
environment, it would mitigate the couch-potato problem. In contrast, future prediction-based curiosity is attracted
to randomness. Our noise ablation, which adds noise to *both* the audio and the visual features, provides some
insight into this.
**R1, R4: Applicability/Generality**: This paper has taken a step forward in terms of real-scenario experiments compared
to prior work, which uses Atari as a standard benchmark. We extend to Habitat, which has many characteristics of the
real world: realistic audio and visual modalities, generated from physical processes, no clear separable sound effects,
and nontrivial associations (i.e. not one-to-one object-to-sound correspondence). We highlight that our method can
perform without direct visibility, in the presence of background noise, and with more than one audio source:
(a) **Visibility**: Direct visibility of an object is not required for association; context should be sufficient. For example,
hearing a microwave sound in a kitchen would be positively associated even if the microwave is out of view.
(b) **Background Noise**: We acknowledge that background noise could be an issue, but it would be so when there is
only background or random noise (as discussed in the above formulation response). If there are foreground audio
and visual signals, we can still learn associations in the presence of noise.
(c) **Multiple Audio Sources**: Multiple objects also render visual prediction hard. It requires object segmentation and
modeling relationships. Similarly, multiple audio sources would require segmentation.
**R1: Issue viewing supplementary material**: Thank you for bringing this to our attention. We cannot change the
format at this time, but you should be able to unzip it with `jar xvf supplementary material.zip`.
**R2: Which baseline from Burda et al.**: Our Burda et al. baseline uses random CNN features, which is stronger than
pixel prediction, as you mentioned. Our method uses the same random CNN features, as shown in Figure 2.
**R2: RND with audio baseline**: We agree that RND would strengthen the audio in baseline ablation. In our preliminary
experiments, RND with audio appears to perform similarly to the RND baseline without audio. We will include this
ablation in the final paper.
**R3: Definition of association**: We think of association as learning shared information between modalities, i.e. the
underlying physical processes that govern these different signals. We do implement this as learning alignment (are two
things cotemporal), but this is not the only way to learn associations. We will modify the text to clarify this.
**R3: Audio padding**: Yes, the padding to 1 second is done to make each sound equal length for feature computation.
**R3: Prior work on mitigating shortcomings**: Thanks for pointing this out and we will add this discussion and context.
Our approach, different from this body of prior work, looks at how multimodal data can mitigate these issues.
**R3: Terminology, prior work, typos, citations**: We are grateful to R3 for their detailed comments and will definitely
incorporate this feedback into the final paper.
**R4: Fixed random initialization**: We use random CNN features to be similar to prior work. This is the same feature
representation as used in our Burda et al. baseline.
**R4: Intrinsic or extrinsic rewards**: The agent only has access to intrinsic rewards, as described at L174-175. Extrinsic
rewards are used only for our evaluation of exploration.

[Meta-Review · NeurIPS 2020]

All four knowledgeable referees support acceptance for the contributions, mainly the idea of multimodal curiosity and its instantiation which can mitigate some issues of certain future-prediction based curiosity approaches and the obtained promising results, and I also recommend acceptance. However, there were important concerns about parts of the paper before the author response and these were clarified in the rebuttal, but that information is currently missing in the paper. The authors should carefully revise the paper to address these concerns (couch potato issue, discussion of environments in which the proposed approach would fail, baselines).